# BNT162b2-boosted immune responses six months after heterologous or homologous ChAdOx1nCoV-19/BNT162b2 vaccination against COVID-19

Georg M. N. Behrens [1,2,3,13] ✉, Joana Barros-Martins [4,13], Anne Cossmann[1,13], Gema Morillas Ramos [1,13], Metodi V. Stankov[1], Ivan Odak[4], Alexandra Dopfer-Jablonka [1,2], Laura Hetzel[1], Miriam Köhler[4], Gwendolyn Patzer[4], Christoph Binz [4], Christiane Ritter[4], Michaela Friedrichsen[4], Christian Schultze-Florey [4,5], Inga Ravens[4], Stefanie Willenzon[4], Anja Bubke[4], Jasmin Ristenpart[4], Anika Janssen[4], George Ssebyatika [6], Verena Krähling[7,8], Günter Bernhardt [4], Markus Hoffmann [9,10], Stefan Pöhlmann[9,10], Thomas Krey [6,11], Berislav Bošnjak [4], Swantje I. Hammerschmidt [4,13] & Reinhold Förster [2,4,12,13] ✉

Heterologous prime/boost vaccination with a vector-based approach (ChAdOx-1nCov-19, ChAd) followed by an mRNA vaccine (e.g. BNT162b2, BNT) has been reported to be superior in inducing protective immunity compared to repeated application of the same vaccine. However, data comparing immunity decline after homologous and heterologous vaccination as well as effects of a third vaccine application after heterologous ChAd/BNT vaccination are lacking. Here we show longitudinal monitoring of ChAd/ChAd ($n = 41$) and ChAd/ BNT ($n = 88$) vaccinated individuals and the impact of a third vaccination with BNT. The third vaccination greatly augments waning anti-spike IgG but results in only moderate increase in spike-specific CD4 + and CD8 + T cell numbers in both groups, compared to cell frequencies already present after the second vaccination in the ChAd/BNT group. More importantly, the third vaccination efficiently restores neutralizing antibody responses against the Alpha, Beta, Gamma, and Delta variants of the virus, but neutralizing activity against the B.1.1.529 (Omicron) variant remains severely impaired. In summary, inferior SARS-CoV-2 specific immune responses following homologous ChAd/ChAd vaccination can be compensated by heterologous BNT vaccination, which might influence the choice of vaccine type for subsequent vaccination boosts.

While the COVID-19 vaccines currently approved by the European Medicines Agency (EMA) and U.S. Food and Drug Administration (FDA) provide high levels of protection against severe illness, the emergence of the Delta and Omicron variants resulted in increasing numbers of breakthrough infections in fully vaccinated individuals[1]. This coincided with evidence of waning immunity in vaccinated individuals[2,3]. Thus, a

third vaccination was proposed to reconstitute immunity and to potentially expand the breadth of immunity against SARS-CoV-2 variants of concern (VoC). Also policy makers have begun to promote a third vaccination not only for vulnerable patients but also to mitigate health-care and economic impact. Real-world data confirm that a third vaccination is effective in preventing COVID-19[4–6]

Concomitant to booster vaccination campaigns, the Omicron variant was identified in South Africa and its emergence was associated with a steep increase in cases and hospitalizations. Omicron has rapidly replaced the highly transmissible Delta variant in many countries[7] with peak infection rates in the first three months in 2022. The S proteins of the Omicron variants BA.1 and BA.2 harbor unusually high numbers of mutations, which increases immune evasion and potentially transmissibility[8–10]. Thus, the Omicron variant developed as a rapidly emerging threat to public health and with the potential to undermine global efforts to control the COVID-19 pandemic.

Heterologous prime-boost strategies appear to offer immunological advantages to strengthen protection against COVID-19 achieved with currently available vaccines. Administration of mRNA vaccines like BNT162b2 (Comirnaty; BNT) after the initial ChAdOx1-nCov-19 (Vaxzevria; ChAd) dose as the second dose of a two-dose regimen was safe and had enhanced immunogenicity compared to homologous ChAd vaccination[11–17]. We have previously reported on the results after homologous and heterologous vaccination after ChAd priming[15].

Here, we aim to assess the effects of a third vaccination after heterologous and homologous prime-boost vaccination on the neutralization of VoCs including Omicron. We show more rapidly waning immunity after two doses of ChAd compared to a first dose of ChAd followed by boosting with BNT, but the application of BNT as a third dose minimises the difference between the two groups. Our data suggest that the choice of vaccine type for subsequent vaccination may influence the magnitude and duration of anti-SARS-CoV-2 immunity.

## Results

In addition to our previously reported findings, we longitudinally monitored immunity after prime-boost COVID-19 vaccine treatment schedules and determined thereafter the impact of BNT booster (Methods). Health care professional vaccinees without previous SARS-CoV-2 infection, who had received ChAd/ChAd or ChAd/BNT, donated further blood four and six months after the second vaccination and about two weeks after the third vaccination. The vaccination and blood collection schedule is depicted in Fig. 1a with additional demographic information (age and sex) in Table 1. A third group of BNT/BNT vaccinees served as an independent control group for serologic analysis only and was monitored for up to nine months (Table 1). As described[15], anti-SARS-CoV-2 spike IgG (anti-S IgG) levels were significantly higher in the ChAd/BNT group short after prime-boost vaccination when compared to the ChAd/ChAd group but declined significantly over time in both groups, with lower anti-S IgG after homologous vaccination prior to the third immunization (Fig. 1b). Note that additional vaccinees after prime-boost vaccination were included in this analysis, which were not yet assessed in our previous report[15]. These data points are shown in color, those previously published in grey.

### Anti-S IgG responses after third vaccination
Following a third immunization, we found greatly increased anti-S IgG responses in both groups. Additional immunization of the homologous ChAd/ChAd immunized group led to a significant 46.9 -fold increase in anti-S IgG ($p < 0.0001$) and 8.0-fold increase in individuals after heterologous ChAd/BNT vaccination ($p < 0.0001$) (Fig. 1b). In both groups, anti-S IgG levels were considerably higher when compared to the situation observed 14 days after the second vaccination. More importantly, the third vaccination diminished previous differences between the heterologous ChAd/BNT and homologous ChAd/

ChAd prime-boost vaccination groups, since anti-S IgG were comparable in both groups after the third vaccination. They were also within the range of triple BNT vaccinated individuals after the third vaccination (Fig. 1c). Please note that in the BNT/BNT/BNT group, samples were collected at different time points than in the other two groups. Because of the non-randomized study design, we assessed age and sex as potential confounders. As depicted in Suppl. Fig. 1A, sex did not have any significant impact on anti-S IgG, but a significant positive correlation to age was found in the ChAd/BNT/BNT, but not in the ChAd/ChAd/BNT group (Suppl. Fig. 1B, C).

### Memory B cell responses after third vaccination
Next, we measured the frequency and phenotype of memory B cells carrying membrane-bound immunoglobulins specific for the Spike protein over time (Methods, Suppl. Fig. 2). Interestingly, the numbers of spike-specific memory B cells generated after prime-boost vaccination gradually increased during the following months with no significant difference between the ChAd/ChAd and the ChAd/BNT group (Fig. 1d). Again, the third vaccination with BNT led to a further and significant expansion of spike-specific memory B cells in both groups (Fig. 1d) in line with increased amounts of spike-specific antibodies, highlighting the impact of the third vaccination for better protection from SARS-CoV-2 infection. Interestingly, 2 weeks after a third immunization with BNT, spike-specific memory B cells were significantly higher in the ChAd/ChAd as compared to the ChAd/BNT prime-boost group (Fig. 1d).

### Neutralization activity against VOCs including Omicron after third vaccination
For the testing of neutralizing activity of antibodies induced by vaccination, we adapted and employed our ELISA-based surrogate virus neutralization test (sVNT) to include Spike proteins of the Omicron variant (Methods)[15,18,19]. For this, we applied sera from vaccinees that had been recently tested for their neutralizing capacity based on vesicular stomatitis virus (VSV) pseudotyped virus neutralization assays (pVNT)[8,18,20]. As for other VoCs[15], we obtained a high degree of correlation between both assays with a R square value of 0.7044 (Suppl. Fig. 3). To further test for the robustness of this assay, we validated the sVNT according to criteria delineated by the European Medicines Agency (Suppl. Figure 4, Suppl. Table 1, Suppl. Note 1).

Using the sVNT assays and consistent with declining anti-S IgG, we confirmed waning neutralizing activity against the Wuhan variant and particularly against the VoCs tested. Whilst the majority of participants had neutralizing antibodies against the Wuhan strain in plasma before the third vaccination, neutralizing antibodies against the Alpha, Beta, Gamma and Delta variants were particularly in the ChAd/ChAd group less frequent or virtually absent (Fig. 2a, b). At 2 weeks after the third immunization, frequencies and titers of neutralizing antibodies against the Wuhan strain increased profoundly in the ChAd/BNT and ChAd/ChAd group with titers reaching values above those after the initial two injections in the latter group (Fig. 2a).

Differences between the vaccination regimens before the third vaccination became even more evident when analyzing the neutralization capacity of vaccine-induced antibodies against the VoCs. In the ChAd/ChAd group, the third immunization profoundly increased neutralization of the Alpha, Beta, Gamma and Delta variants (Fig. 2b), which was low for Alpha and Delta after prime/boost and virtually absent for Beta and Gamma. Whilst initial ChAd/BNT immunization had induced neutralizing antibodies at high levels against all analyzed VoCs except for Beta and Gamma after ChAd/ChAd vaccination, the following decline was more than restored by the third vaccination (Fig. 2a, b). In fact, nearly all BNT-boosted ChAd/BNT vaccinees had efficient neutralizing activity against Alpha, Beta, Gamma, and Delta and titers were mostly above those identified after the second vaccination. Importantly, the neutralization capacity against the Omicron

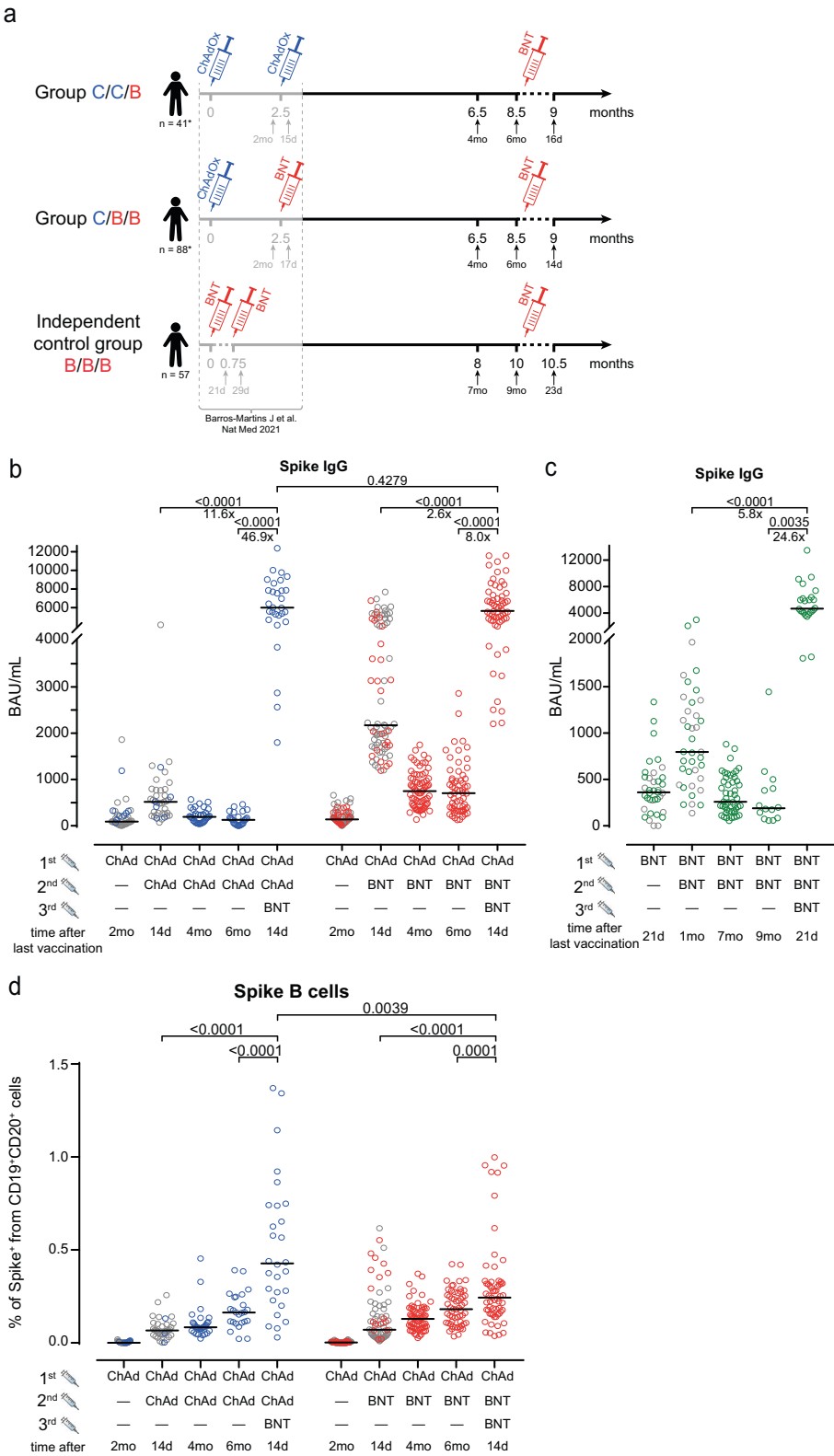

**Fig. 1 | Participant recruitment schemes and humoral immune response.**
**a** Participant recruitment and vaccination and blood sampling scheme. C, ChAd; B, BNT. *Note that additional participants, which had not passed the analysis time point for the previous analyses[15,19] were included in the current study about the third vaccination. **b** A third heterologous or **c**, a third homologous immunization with BNT induce strong increases in anti-S1 IgGs. **d** A third heterologous immunization with BNT leads to increased frequencies of S-specific memory B cells. Statistics: **b**–**d** Mixed effect analysis followed by Sidak's multiple comparison test (within groups) and unpaired two-sided *t* test with Welch's correction (between groups). The symbols depicted in grey have been published before[15,19]. Source data are provided as a Source Data file.

**Table 1 | Demographic data at third vaccination and median time in days months since last vaccination as indicated for the five blood collection time points of the three vaccination groups**

| | Mean age, years (range) | Sex, m/f (%) | Median (IQR) days [month] past last vaccination | | | | |
|---|---|---|---|---|---|---|---|
| | | | After 1st vaccination | After 2nd vaccination | | | After 3rd vaccination |
| ChAd ChAd BNT | 40 (21–64) | 14/27 (34/66) | 68 (12.75) [2] | 15 (4) | 119 (13) [4] | 196 (5) [6] | 16 (6.5) |
| ChAd BNT BNT | 37 (19–61) | 17/65 (21/79) | 70 (8) [2] | 17 (5) | 117 (13) [4] | 195 (8) [6] | 14 (2) |
| BNT BNT BNT | 42 (23–63) | 21/36 (38/62) | 20 (1.25) | 29 (8.25) | 211 (9) [7] | 267 (22.5) [9] | 23 (10.25) |

variant was virtually absent 14 days after the second vaccination in the ChAd/ChAd group, while anti-Omicron titers could be revealed at that time in 36/78 (46%) ChAd/BNT vaccinees (Fig. 2a). In contrast, 14 days after the third vaccination Omicron-neutralizing antibodies were present in 28/29 (97%) and 55/58 (95%) of vaccinees in the ChAd/ChAd and ChAd/BNT group, respectively (Fig. 2a). We obtained very similar results after BNT booster in BNT/BNT vaccinated individuals (Suppl. Fig. 5). Altogether, these data indicate that the third immunization led to an increase of neutralizing antibodies in both vaccination groups against all tested VoCs including Omicron.

### Anti-S T cell responses after third vaccination

Finally, we also analyzed frequencies and phenotypes of spike-specific T cells (Methods, Suppl. Figs. 6 and 7). We quantified numbers of spike-specific T cells as the sum of all cells producing IFN-γ or TNF-α as described previously[15]. The frequencies of spike-specific CD4+ and CD8+ T cells in blood samples collected after the second vaccination were significantly higher in the ChAd/BNT than in the ChAd/ChAd group (Fig. 3a, b). Both cell populations declined over time after heterologous immunization, while they remained at low levels after homologous vaccination. Whilst spike-specific CD4+ T cells declined to frequencies similar to individuals after homologous ChAd/ChAd vaccination (Fig. 3a), spike-specific CD8+ T cells remained above the frequencies of the heterologous vaccinated group (Fig. 3b).

More interestingly, a third immunization with BNT in the ChAd/ChAd group significantly raised numbers of spike-specific CD4+ T cells above levels observed after the second vaccination (Fig. 3a). In contrast, a third vaccination of the ChAd/BNT group only regained spike-specific CD4+ T cell to levels present after the second vaccination (Fig. 3b). Similarly, a third vaccination with with BNT did not result in an expansion of spike-specific CD8+ T cells above levels observed after second vaccination in ChAd/BNT vaccinees, but did so in ChAd/ChAd vaccinated individuals (Fig. 3b). Like for spike-specific CD4+ T cells, raised numbers in spike-specific IFN-γ-producing T cells in the ChAd/ChAd as well as the ChAd/BNT group after the additional BNT vaccination was confirmed by cytokine measurement in supernatants after SARS-CoV-2 spike peptide stimulation (Fig. 3c). Again, the third vaccination did not further increase spike-specific IFN-γ-producing T cells in ChAd/BNT vaccinated subjects above levels already after the second vaccination.

### Discussion

The third vaccination potently increased anti-S IgG in all heterologous and homologous vaccinated individuals tested and this rise was accompanied by further strengthened neutralizing capacity against the Wuhan variant and Alpha, Beta, Gamma, and Delta. We obtained mean antibody levels after initial homologous ChAd/ChAd or heterologous ChAd/BNT vaccination followed by a third BNT vaccination of around 6.000 BAU/mL or after triple BNT vaccination of about

4.000 BAU/mL. These results are within the range of other reports after triple antigen exposure either by SARS-CoV-2 infection and subsequent vaccination or triple vaccination. Using the same assay, Wratil PR et al. found anti-S levels of 4.000 to 6.000 BAU/mL in triple vaccinated individuals or convalescent persons with double vaccination after COVID-19[21]. Similarly, others reported about 2.000–3.000 BAU/mL after triple BNT vaccination based on a different ELISA[22], or obtained anti-S IgG levels of about 3.000 BAU/mL in individuals after ChAd/Chad/BNT vaccination[23], both using different assays to those employed here.

These data corroborate findings after homologous vaccination[9,10,24–31], reports about vaccine protection[32], and support current recommendations by the European Medicines Agency (EMA) and the European Centre for Disease Prevention and Control (ECDC). A third vaccination was particularly important to induce neutralizing antibodies against Omicron. While two weeks after the second vaccination, neutralizing antibodies were detectable in none (0/40) of the ChAd/ChAd vaccinees and in 46% (36/78) in the ChAd/BNT group, the third vaccine application induced neutralizing antibodies in 83/87 individuals with median titers increasing 9-fold confirming results of others[10,21,27,31]. However, neutralization of the Omicron variant was absent before second booster and remained clearly inferior thereafter, irrespective of the previous vaccination scheme. Considering the kinetics of waning neutralizing antibodies against the other VoCs after the second vaccination, we expect remaining neutralization against Omicron to vanish rapidly in the majority of vaccinees despite persisting high anti-S IgG concentrations. Recent reports support this assumption, reporting waning immunity after the third vaccination[32–34] and showing that the protection against confirmed infection with the Omicron variant reaches a maximum in the fourth week after 4th immunization compared to those that received 3 doses[35]. Since the current vaccines show low efficacy in preventing infection with the Omicron variants, it will be interesting to see, whether Omicron-adapted vaccines will show better efficacy preventing infection. Initial results from animal studies are unfortunately not particularly promising. Third vaccinations with Omicron-adapted vaccines after standard prime-boost immunization lead to only limited differences in efficacy measured in mice[36] and boosting non-human primates with either the mRNA-1273 or an mRNA-Omicron vaccine elicit similar levels of protection upon challenge with SARS-CoV-2 Omicron[37].

The third vaccination in our study made up for the absent rise in spike-specific T cell responses after homologous ChAd/ChAd vaccination, a finding that confirms and extends reports by others that the ChAd vaccine does not boost cellular responses after a second application[12,38]. Although the relative role of T cell immunity remains unclear, spike-specific T memory cells are probably of great importance for protection against severe COVID-19, hospitalization, and death. Studies investigating CD4+ and CD8+ T cells in vaccinated and convalescent individuals revealed a high degree of preservation of T

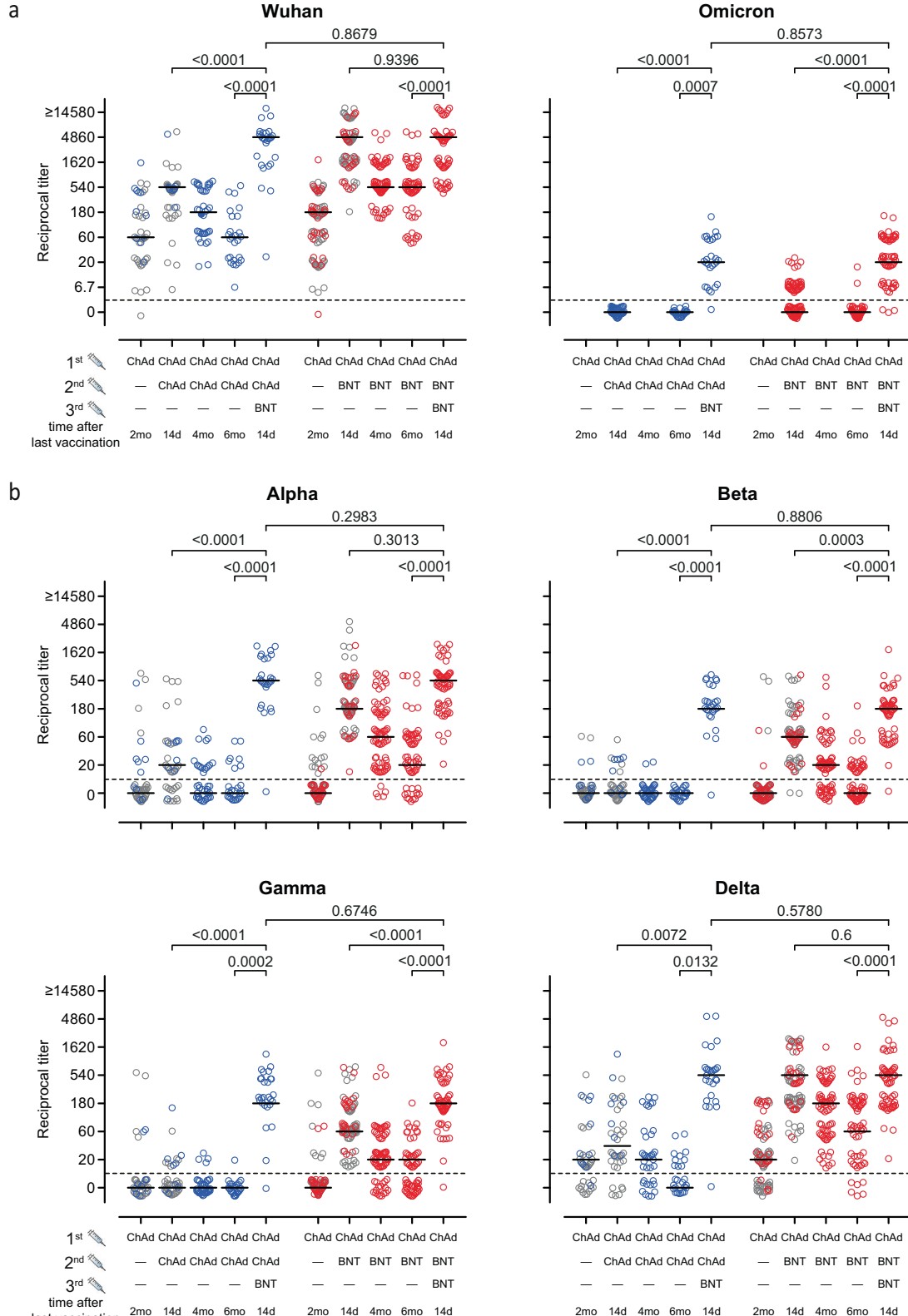

**Fig. 2 | Heterologous vaccination induces neutralizing antibodies.** Heterologous ChAd/BNT/BNT or ChAd/ChAd/BNT vaccination induces neutralizing antibodies against (**a**), Wuhan and B.1.1.529 (Omicron) as well as **b**, B.1.1.7 (Alpha), B.1.351 (Beta), P.1 (B.1.1.28.1; Gamma), and B.1.617.2 (Delta) SARS-CoV-2-S variants measured using the sVNT. For better visualization of identical titer values, data were randomly and proportionally adjusted closely around the precise titer results. The dotted line represents the lower limit of detection. **a**, **b**. Mixed effect analysis followed by Sidak's multiple comparison test (within groups) and unpaired two-sided *t* test with Welch's correction (between groups). The symbols depicted in grey have been published before[15,19]. Source data are provided as a Source Data file.

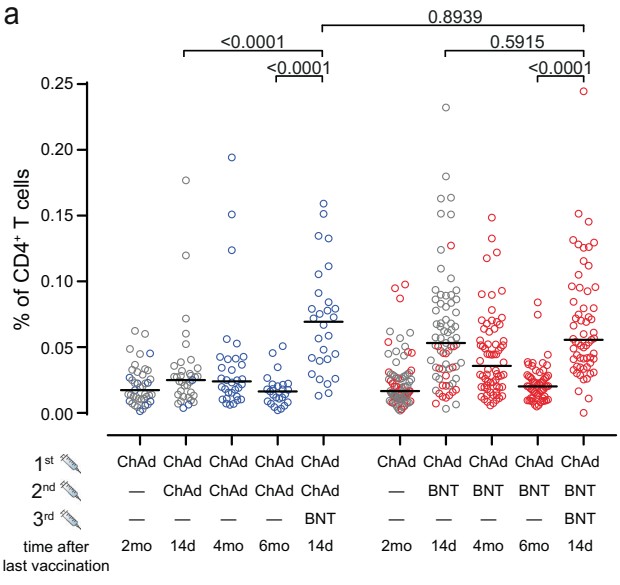

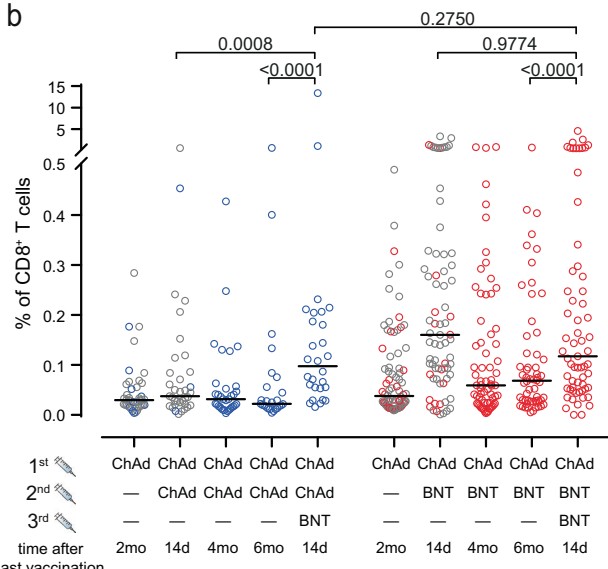

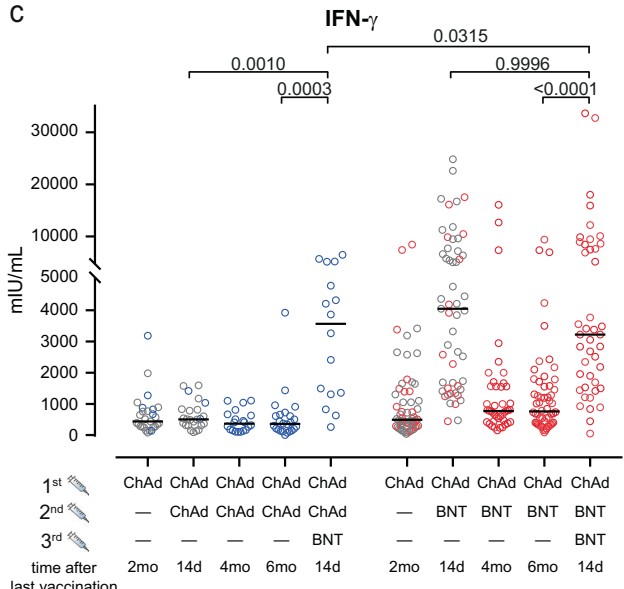

**Fig. 3 | Heterologous vaccination induces CD4 and CD8 T cell responses.**
**a** Heterologous ChAd/BNT/BNT or ChAd/ChAd/BNT vaccination increased total percentage of cytokine-secreting CD4[+] (**a**) and CD8[+] (**b**) T cells. We calculated the total number of cytokine secreting cells as the sum of IFN-γ + TNF-α − , IFN-γ + TNF-α + and IFN-γ − TNF-α + cells in the gates indicated in Extended Data Fig. 6. **c**, IFN-γ concentration in full blood supernatants after stimulation with SARS-CoV-2 S1 domain for 20–24 h measured by IGRA (Euroimmun). **a**–**c** Mixed effect analysis followed by Sidak's multiple comparison test (within groups) and unpaired two-sided *t* test with Welch's correction (between groups). The symbols depicted in grey had been published before[15,19]. Source data are provided as a Source Data file.

cell epitopes between the ancestral strain and Omicron[30,39–41]. Our data reveal that a third immunization with BNT has a limited effect on the expansion of spike-specific CD8[+] cells in the ChAd/BNT group and suggest that novel vaccines and vaccine schedules should be explored for further strengthening of adaptive cellular immunity against SARS-CoV-2 and its variants[42]. Such vaccines should also aim to target other structural viral proteins including nucleocapsid and membrane proteins, which are less likely to be able to escape from capable immune recognition.

## Methods

### Participants

Participants for this analysis were from the COVID-19 Contact (CoCo) Study (German Clinical Trial Registry, DRKS00021152), an ongoing, prospective observational study monitoring anti-SARS-CoV-2 IgG immunoglobulin and immune responses in health care professionals

(HCP) at Hannover Medical School and individuals with a potential contact to SARS-CoV-2[43,44]. An amendment from Dec 2020 allowed us to study the immune responses after COVID-19 vaccination. We followed the study cohort described previously[15] after heterologous ChAd/BNT or homologous ChAd/ChAd and BNT/BNT vaccination. Data collection including questionnaires and lab assessment, was done in Excel 2016. Scheduling appointments for a third booster vaccination with BNT was coordinated by an independent vaccination team according to vaccine availability.

One individual with previous SARS-CoV-2 infection as determined by positive anti-SARS-CoV-2 NCP IgG before vaccinations were excluded from this analysis. Two additional individuals each in the ChAd/ChAd and ChAd/BNT group developed anti-SARS-CoV-2 NCP IgG after prime/boost vaccination and were excluded from follow-up analysis. Demographics (sex and age) are depicted in Table 1. After blood collection, we separated plasma from EDTA or lithium heparin blood

**Table 2 | Immunization scheme and n = biological independent samples analyzed in the assays indicated**

| | | ChAd | ChAd | ChAd | ChAd | ChAd | ChAd | ChAd | ChAd | ChAd | ChAd | BNT | BNT | BNT | BNT | BNT |
|---|---|---|---|---|---|---|---|---|---|---|---|---|---|---|---|---|
| 1st vaccination | | ChAd | ChAd | ChAd | ChAd | ChAd | ChAd | ChAd | ChAd | ChAd | ChAd | BNT | BNT | BNT | BNT | BNT |
| 2nd vaccination | | ----- | ChAd | ChAd | ChAd | ChAd | ----- | BNT | BNT | BNT | BNT | ----- | BNT | BNT | BNT | BNT |
| 3rd vaccination | | ----- | ----- | ----- | ----- | BNT | ----- | ----- | ----- | ----- | BNT | ----- | ----- | ----- | ----- | BNT |
| Time after last vaccination | | 2 mo | 14d | 4 mo | 6mo | 14d | 2 mo | 14d | 4 mo | 6mo | 14d | 21d | 1mo | 7 mo | 9 mo | 21d |
| Anti-S1-IgG | Fig. 1b, c | 41 | 39 | 37 | 26 | 29 | 80 | 80 | 73 | 61 | 61 | 37 | 38 | 49 | 14 | 24 |
| Spike-specific B cells | Fig. 1d | 41 | 35 | 32 | 25 | 28 | 83 | 74 | 68 | 59 | 61 | n.d. | n.d. | n.d. | n.d. | n.d. |
| sVNT Wuhan | Fig. 2a, Suppl. Fig. 5a | 40 | 39 | 37 | 26 | 27 | 82 | 79 | 73 | 60 | 59 | 34 | 47 | 33 | 48 | 24 |
| sVNT Alpha | Fig. 2b, Suppl. Fig. 5b | 40 | 39 | 37 | 26 | 27 | 82 | 79 | 73 | 60 | 59 | 34 | 47 | 33 | 48 | 24 |
| sVNT Beta | Fig. 2b, Suppl. Fig. 5b | 40 | 39 | 37 | 26 | 27 | 82 | 79 | 73 | 60 | 59 | 34 | 47 | 33 | 48 | 24 |
| sVNT Gamma | Fig. 2b, Suppl. Fig. 5b | 40 | 39 | 37 | 26 | 27 | 82 | 79 | 73 | 60 | 59 | 34 | 47 | 33 | 48 | 24 |
| sVNT Delta | Fig. 2a, Suppl. Fig. 5a | 40 | 39 | 37 | 26 | 27 | 82 | 79 | 73 | 60 | 59 | 34 | 47 | 33 | 48 | 24 |
| sVNT Omicron | Fig. 2a, Suppl. Fig. 5a | n.d. | 39 | n.d. | 26 | 27 | n.d. | 79 | n.d. | 60 | 59 | n.d. | 44 | n.d. | 48 | 24 |
| Spike-specific CD4 cells | Fig. 3a, Suppl. Fig. 7 | 41 | 35 | 32 | 25 | 28 | 82 | 74 | 68 | 59 | 61 | n.d. | n.d. | n.d. | n.d. | n.d. |
| Spike-specific CD8 cells | Fig. 3b, Suppl. Fig. 7 | 41 | 34 | 32 | 25 | 28 | 82 | 67 | 68 | 59 | 63 | n.d. | n.d. | n.d. | n.d. | n.d. |
| IGRA | Fig. 3c | 27 | 24 | 20 | 24 | 16 | 60 | 53 | 42 | 56 | 43 | n.d. | n.d. | n.d. | n.d. | n.d. |

*n.d* not determined.

(S-Monovette, Sarstedt) and stored it at −80 °C until use. We used full blood or isolated PBMCs from whole blood samples by Ficoll gradient centrifugation and for stimulation with SARS-CoV-2 peptide pools. The number of biological independent samples analyzed in the different assays are indicated in Table 2.

### Pseudotyped virus neutralization assay (pVNT)

pVNTs were performed at the Infection Biology Unit of the German Primate Center in Göttingen as described recently[20]. Briefly, the rhabdoviral pseudotyped particles were produced in 293 T cells transfected to express the desired SARS-CoV-2-S variant inoculated with VSV*DG-FLuc, a replication-deficient VSV vector that encodes for enhanced green fluorescent protein and firefly luciferase (FLuc) instead of VSV-G protein (kindly provided by Gert Zimmer, Institute of Virology and Immunology, Mittelhäusern, Switzerland). Produced pseudoparticles were collected, cleared from cellular debris by centrifugation, and stored at −80 °C until used. For neutralization experiments, equal volumes of pseudotyped particles and heat-inactivated (56 °C, 30 min) plasma samples serially diluted in a culture medium were mixed and incubated for 30 min at 37 °C. Afterwards, the samples together with non-plasma-exposed pseudotyped particles, were used for transduction experiments. The assay was performed in 96-well plates in which Vero cells were inoculated with the respective pseudotyped particles/plasma mixtures. The transduction efficacy was analyzed at 16-18 hr post-inoculation by measuring FLuc activity in lysed cells (Cell culture lysis reagent, Promega) using a commercial substrate (Beetle-Juice, PJK) and a plate luminometer (Hidex Sense Plate Reader, Hidex) with the Hidex Sense Microplate Reader Software (version 0.5.41.0).

### Serology

We measured SARS-CoV-2 IgG by quantitative ELISA (anti-SARS-CoV-2 S1 Spike protein domain/receptor binding domain IgG SARS-CoV-2-QuantiVac, EI 2606-9601-10 G, Euroimmun, Lübeck, Germany) according to the manufacturer's instructions (dilution up to 1:4000). We provide anti-S1 concentrations expressed as RU/mL as assessed from a calibration curve with values above 11 RU/mL defined as positive. These values can be converted in binding antibody units (BAU/mL) by multiplying RU/mL by 3.2. We performed anti SARS-CoV-2 nucleocapsid (NCP) IgG measurements according to the manufacturer's instructions (Euroimmun, Lübeck, Germany). We used an AESKU.READER (AESKU.GROUP, Wendelsheim, Germany) and the Gen5 2.01 Software for analysis.

### Surrogate virus neutralization assay (sVNT) for SARS-CoV-2 variants

To determine neutralizing antibodies against the Wuhan-Spike, the B.1.1.7-Spike (Alpha), the B.1.351-Spike (Beta), the P.1-Spike (B.1.1.28.1; Gamma), the B.1.617.2 (Delta), and the B.1.1.529 BA.1 (Omicron BA.1) variants of SARS-CoV-2-S in plasma, we modified our recently established surrogate virus neutralization test (sVNT)[18,19]. In this assay, the soluble receptor for SARS-CoV-2, ACE2, is bound to 96-well-plates to which different purified tagged receptor binding domains (RBDs) of the Spike-protein of SARS-CoV-2 can bind once added to the assay. Binding is further revealed by an anti-tag peroxidase-labelled antibody and colorimetric quantification. Pre-incubation of the Spike-protein with serum or plasma of convalescent patients or vaccinees prevents subsequent binding to ACE2 to various degrees, depending on the amount of neutralizing antibodies present. In detail, MaxiSorp 96 F plates (Nunc) were coated with recombinant soluble hACE2-Fc(IgG1) protein at 300 ng per well in 50 μL coating buffer (30 mM Na2CO3, 70 mM NaHCO3, pH 9.6) at 4 °C overnight. After blocking with hACE2-Fc(IgG1), plates were washed with phosphate-buffered saline, 0.05% Tween-20 (PBST), and blocked with BD OptEIA Assay Diluent for 1.5 h at 37 °C. In the meantime, plasma samples were serially diluted threefold starting at 1:6,7 or 1:20 and then pre-incubated for 1 h at 37 °C with 1.5 ng recombinant SARS-CoV-2 Spike RBD of either the Wuhan strain (Trenzyme), the B.1.1.7 variant (N501Y; Alpha), the B.1.351 variant (K417N, E484K, N501Y; Beta), the P.1 variant (K417T, E484K, N501Y; Gamma), the B.1.617.2 variant (L452R,T478K) or the B.1.1.529 BA.1 variant (G339D, S371L, S373P, S375F, K417N, N440K, G446S, S477N, T478K, E484A, Q493R, G496S, Q498R, N501Y, Y505H) (the latter five products from SinoBiological), all with a C-terminal His-Tag. BD OptEIA Assay Diluent was used for preparing plasma samples as well as RBD dilutions. After pre-incubation with SARS-CoV-2 Spike RBDs, plasma samples were given onto the hACE2-coated MaxiSorp ELISA plates for 1 h at 37 °C. SARS-CoV-2 Spike RBDs pre-incubated with buffer only served as negative controls for inhibition. Plates were washed three times with PBST and incubated with an HRP-conjugated anti-His-tag antibody (clone HIS 3D5, provided by Helmholtz Zentrum München) for 1 h at 37 °C. Unbound antibody was removed by six washes with PBST. A colorimetric signal was developed on the enzymatic reaction of HRP with the chromogenic substrate 3,3',5,5'-tetramethylbenzidine (BD OptEIA TMB Substrate Reagent Set). An equal volume of 0.2 M H2SO4 was added to stop the reaction, and the absorbance readings at 450 nm and 570 nm were acquired using a SpectraMax iD3 microplate reader (Molecular Devices) using SoftMAX Pro v7.03 software. For

each well, the percent inhibition was calculated from optical density (OD) values after subtraction of background values as: Inhibition (%) = (1 − Sample OD value/Average SARS-CoV-2 S RBD OD value) × 100. Neutralizing sVNT titers were determined as the dilution with binding reduction >mean + 2 SD of values from a plasma pool consisting of three pre-pandemic plasma samples.

### SARS-CoV-2 protein peptide pools

We ordered 15 amino acid (aa) long and 10 aa overlapping peptide pools spanning the whole length of Wuhan SARS-CoV2-Spike (-S) (total 253 peptides) protein from GenScript. All lyophilized peptides were synthesized at >95% purity and reconstituted at a stock concentration of 50 mg/mL in DMSO (Sigma-Aldrich), except for 9 SARS-CoV2-S overlapping peptides (number 24, 190, 191, 225, 226, 234, 244, 245, and 246) that were dissolved at 25 mg/mL due to solubility issues. All peptides in DMSO stocks were stored at −80 °C until used.

### T cell re-stimulation assay

PBMCs, isolated using a Ficoll gradient, were re-suspended at concentration of $20 \times 10^6$ cells/mL in complete RPMI medium [RPMI 1640 (Gibco) supplemented with 10% FBS (GE Healthcare Life Sciences, Logan, UT), 1 mM sodium pyruvate, 50 µM β-mercaptoethanol, 1% streptomycin/penicillin (all Gibco)]. For stimulation, cells were diluted with equal volume of the S-protein. The peptide pool was prepared in complete RPMI containing brefeldin A (Sigma-Aldrich) at final concentration of 10 µg/mL. In the final mixture each peptide had concentration of 2 µg (~1.2 nmol)/mL, except for SARS-CoV2-S peptides number 24, 190, 191, 225, 226, 234, 244, 245, and 246, which were used at final concentration of 1 µg/mL due to solubility issues. As a negative control, we stimulated the cells with DMSO, in volume corresponding to DMSO amount in peptide pools (equaling to 5 % DMSO in final medium volume) Extended Fig. 5. In each experiment, we used cells stimulated with Phorbol-12-myristate-13-acetate (PMA; Calbiochem) and ionomycin (Invitrogen) at final concentration of 50 ng/mL and 1500 ng/mL, respectively, as an internal positive control. Cells were then incubated for 12-16 h at 37 °C, 5% $CO_2$. After washing, cells were resuspended in MACS buffer (PBS supplemented with 3% FBS and 2 mM EDTA). Non-specific antibody binding was blocked by incubating samples with 10% mouse serum at °C for 15 min. Next, without washing, an antibody mix of anti-CD3-APC-Fire810 (SK7; # 344858; Lot # B331674; Biolegend; 1:50), anti-CD4-BUV563 (RPA-T4; #741353; Lot # 0295029; BD Biosciences; 1:200), anti-CD8-SparkBlue550 (SK1; #344760; Lot #B326454; Biolegend; 1:200), anti-CD45RA-BUV395 (HI100, #740298, Lot # 0295008/1270969; BD Biosciences; 1:200), anti-CCR7-BV785 (G043H7; #353230; Lot # B335328; Biolegend; 1:50) and Zombie Yellow™ Fixable Viability Kit (#423104; Lot # B272131; BioLegend; 1:400) was added. After staining for 20 min at RT, cells were washed before they were fixed and permeabilized (#554714; BD Biosciences) according to the manufacturers' protocol. Next, intracellular cytokines were stained using anti-IFN-γ-PE-Cy7 (B27; #506518; Lot # B326674; Biolegend; 1:100), anti-TNF-α-AF700 (Mab11; #502928; Lot # B337546; Biolegend; 1:50) for 45 min on RT. Excess antibodies were washed away, and cells were then acquired on Cytek Aurora spectral flow cytometer (Cytek) equipped with five lasers operating on 355 nm, 405 nm, 488 nm, 561 nm, and 640 nm (for gating strategy, see Suppl. Fig. 6). All flow cytometry data were acquired using SpectroFlo v2.2.0 (Cytek) and analyzed by FCS Express V7 (Denovo).

### Flow cytometric analysis of spike-specific B cells

Total leukocytes were isolated from whole blood using erythrolysis in 0.83% ammonium chloride solution. Isolated cells were then washed, counted and resuspended in PBS and stained for 20 min on RT with an antibody mix containing antibodies listed in Suppl. Fig. 2A together with Spike-mNEONGreen fusion protein (5 µg per reaction). After one wash, samples were acquired on a spectral flow cytometer, and the data were analyzed as described above (for gating strategy, see Suppl. Fig. 2B).

### Quantification of IFN-γ release

0.5 mL full blood were stimulated with the manufacturer's selected parts of the SARS-CoV-2 S1 domain of the Spike Protein for a period of 20-24 h (ET 2606-3003, SARS-CoV-2 Interferon Gamma Release Assay, IGRA (Euroimmun, Lübeck, Germany). We carried out negative and positive controls according to the manufacturer's instruction and measured IFN-γ using an ELISA (EQ 6841-9601, Euroimmun, Lübeck, Germany). For analysis, we used an AESKU.READER (AESKU.GROUP, Wendelsheim, Germany) and the Gen5 2.01 Software.

### Statistics

Statistical analysis was done using GraphPad Prism 8.4 or 9.0 (GraphPad Software, USA) and SPSS 20.0.0 (IBM SPSS Statistics, USA). For comparison of levels of Spike-specific IgG levels, as well as for comparison of percentages of cytokine-secreting T cells, for comparision of frequencies of Spike-specific B cells, or cytokine concentrations in the IGRA assay and sVNT values, we used mixed-effect analysis with Sidak's multiple comparison paired *t*-test (within groups) or unpaired *t* test with Welch correction (between groups). Percentages of cytokine secreting T cells were log-transformed prior to comparison. Differences were considered significant if $p < 0.05$. The correlation between sVNT and pVNT or anti-S IgG and age values was calculated using a single linear regression analysis.

### Ethics committee approval

The CoCo Study and the analysis conducted for this article were approved by the Internal Review Board of Hannover Medical School (institutional review board no. 8973_BO-K_2020, amendment Dec 2020). All study participants gave written informed consent and received no compensation.

### Inclusion & ethics statement

For this research, local researchers were included throughout the research process including study design, study implementation, data ownership, and authorship. The research is locally relevant and has been determined in collaboration with local partners. All roles and responsibilities for e.g. sample collection and handling, respective immunological studies, data curation and analysis were agreed amongst collaborators ahead of the research. We have considered local and regional research relevant to our study in the citations.

### Reporting summary

Further information on research design is available in the Nature Research Reporting Summary linked to this article.

## Data availability

The data generated in this study are provided as Source Data files in conjunction to this manuscript. All requests for raw and analyzed data that underlie the results reported in this article will be reviewed within four weeks by the CoCo Study Team, Hannover Medical School (cocostudie@mh-hannover.de) to determine whether the request is subject to confidentiality and data protection obligations. Data that can be shared will be released via a material transfer agreement. Source data are provided with this paper.

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

## Acknowledgements

This work was supported by the German Center for Infection Research TTU 01.938 (grant no 80018019238 to G.M.N.B and R.F.), and TTU 04.820 to G.M.N.B., by Deutsche Forschungsgemeinschaft, (DFG, German Research Foundation) Excellence Strategy EXC 2155 "RESIST" (Project ID39087428 to R.F.), by funds of the State of Lower Saxony (14-76103-184 CORONA-11/20 to R.F.), by funds of the BMBF (NaFoUniMedCovid19 FKZ: 01KX2021; Projects B-FAST to R.F.) and Deutsche Forschungsgemeinschaft, SFB 900/3 (Projects B1, 158989968 to R.F.), and the European Regional Development Fund (Defeat Corona, ZW7-8515131 to A.D-J. and G.M.N.B. and ZW7-85151373 to G.M.N.B.). We thank the CoCo Study participants for their support and the entire CoCo study team for help. We would like to thank Luis Manthey, Annika Heidemann, Till Redeker, Madeleine Rommel, Christian Sturm, Marie Mikuteit, Jacqueline Niewolik, Ruth Sikora, Janine Topal, Kerstin Sträche, Birgit Heinisch, Michael Stephan, Mariel Nöhre, Simone Müller, Olivera Dragicevic, Kim Do Thi Hoang, Amy Kempf, and Inga Nehlmeier for technical and logistical support.

## Author contributions

Study design: G.M.N.B and R.F. Data collection: J.B.-M., S.I.H., A.C., I.O., M.V.S., G.M.R., A.D-J., L.H., M.K, G.P., C.B., C.R., M.F., C. S.-F., I.R., S.W., A.B., J.M., J.R., A.J., G.S., G.B., J.M., M.H., S.P., T.K., V.K. Data analysis: G.M.N.B, J.B.-M., S.I.H., A.C., I.O., G.M.R., M.H., B.B, M.V.S., V.K. Data interpretation: R.F., G.M.N.B. Writing: G.M.N.B., R.F. with comments from all authors.

## Funding

## Competing interests

The authors declare no competing interests.

## Additional information

[1]Department for Rheumatology and Immunology, Hannover Medical School, 30625 Hannover, Germany. [2]German Center for Infection Research (DZIF), Partner Site Hannover-Braunschweig, 30625 Hannover, Germany. [3]CiiM, Centre for Individualized Infection Medicine, Hannover, Germany. [4]Institute of Immunology, Hannover Medical School, 30625 Hannover, Germany. [5]Department of Hematology, Hemostasis, Oncology and Stem-Cell Transplantation, Hannover Medical School, 30625 Hannover, Germany. [6]Institute of Biochemistry, University of Lübeck, 23562 Lübeck, Germany. [7]Institute of Virology, Philipps University Marburg, 35043 Marburg, Germany. [8]German Center for Infection Research (DZIF), Partner Site Gießen-Marburg-Langen, 35043 Marburg, Germany. [9]Infection Biology Unit, German Primate Center, 37077 Göttingen, Germany. [10]Faculty of Biology and Psychology, Georg-August-University Göttingen, 37073 Göttingen, Germany. [11]German Center for Infection Research (DZIF), Partner Site Hannover-Braunschweig and Partner Site Hamburg-Lübeck-Borstel-Riems, Hamburg, Germany. [12]Cluster of Excellence RESIST (EXC 2155), Hannover Medical School, 30625 Hannover, Germany. [13]These authors contributed equally: Georg M. N. Behrens, Joana Barros-Martins, Anne Cossmann, Gema Morillas Ramos, Swantje I. Hammerschmidt, Reinhold Förster. ✉e-mail: Behrens.georg@mh-hannover.de; foerster.reinhold@mh-hannover.de

