## [Peer Review File · Nature Communications]

BNT162b2-boosted immune responses six months after heterologous or homologous ChAdOx1nCoV-19/BNT162b2 vaccination against COVID-19REVIEWER COMMENTS

Reviewer #1 (Remarks to the Author):

This manuscript investigated the effects of a 2nd booster (3rd dose) with BNT162b2 mRNA vaccine in individuals previously primed with two doses of the ChAd vaccine or a prime and one boost with the BNT mRNA vaccine. With the development of multiple COVID-19 vaccines and the need for worldwide distribution, vaccine mixing is inevitable and determining the ability of vaccine combinations to induce neutralizing antibody against new VOCs including Omicron is critical. This is a well-designed study comparing ChAd/ChAd/BNT to ChAd/BNT/BNT combinations and including a BNT only control and the methods which includes the analysis of both antibody and T cell responses and statistical methods are sound. However, there are some major and moderate issues that reduce confidence in the findings and interpretations.

Major:

1. A primary conclusion from this study is that an Omicron specific vaccine will be needed. This is based on data showing that after the 2nd booster dose, 31 or 47% of the vaccinees in both test groups failed to develop detectable neutralization activity against Omicron. A major concern with this conclusion is that the results in the control group (BNT/BNT/BNT) similarly show a relatively high rate of non-responders to Omicron after the 3rd dose but this is not consistent with previous reports which reported 100% of subjects developing neutralizing antibody after the 3rd dose in a BNT/BNT/BNT immunization regiment (<https://www.nature.com/articles/s41591-021-01676-0>). The authors should provide possible reasons for this discrepancy.
2. Immune responses measured after the 2nd booster are limited to only one time points at 2 weeks after the dose. A single measurement after the immunization reduces confidence in then conclusions. If the samples are available, the authors should include data at a later timepoint to validate the findings at day 14 and support their conclusion that "we expect the neutralization against Omicron to vanish rapidly.."
3. The discussion does not adequately compare the results from this study to similar studies that have already been published investigating the impact of homologous boosting with a 3rd dose of mRNA vaccine on breadth of neutralization against VOCs including Omicron. See link above and also: <https://www.nejm.org/doi/full/10.1056/NEJMc2119912>

Minor:

1. Some of the conclusions are based on % of subjects that developed detectable neutralizing antibody vs the VoCs including Omicron after the 2nd booster but these findings are not that clear from the figures. With regard to number of vaccinees that have detectable neutralizing antibody vs not, this could be improved if the authors insert a threshold or lower limit of detection line in each graph so it is more apparent which data points are above vs below. It would also help if the author indicated what % of subjects are above the threshold for each timepoint and group somewhere in the figure.
2. The authors established a virus neutralization test (sVNT) that positively correlates with neutralization detected via well-established pseudotyped virus neutralization test (pVNT). Does this assay have a lower or higher limit of detection? This should be made clear in the methods or results section.
3. In describing the results, it is not always clear if the authors are referring to the responses after the first booster or second booster. It would be helpful if the same terms (i.e. after first booster vs after second booster) are consistently used throughout the manuscript.
4. In the methods, the authors should include information on the virus specificity (Wu-1?) of the peptide pools they used for stimulation in the T cell assays.

Reviewer #2 (Remarks to the Author):

The authors summarized longitudinal immune responses after ChAd/ChAd and ChAd/BNT, as well as immune responses 14 days after a BNT boost following the primary series against multiple variants of SARS-CoV-2. The paper is very clearly written and easy to follow. Here are few comments.

Major:

1. The conclusion about the sVNT assay being a reliable tool to quantitatively assess neutralization seems weak. Such statement would need to have a formal validation of the assay including parameters such as accuracy, precision, range, limits of detection or quantification etc. Why an R^2 value of 0.7044 supports this?
2. Given that the correlates of the Moderna vaccine have been reported (Gilbert et al. 2022), it's ideal if the magnitudes of the reported neutralization titers could be converted to the WHO international units for a comparison to other vaccine regimens.
3. Have the authors tried to adjust for any potential confounding factors in the comparisons between the two groups given that they were not randomized groups. This will be helpful at least as a supportive analysis.

Minor:

1. The word "efficacy" in "Assessing the *efficacy* of a second boost vaccination" isn't appropriate as no clinical outcomes are assessed.

Reviewer #3 (Remarks to the Author):

Thank you for the opportunity to review. In this manuscript, the authors follow up their prior work (Barros-Martins, et al in Nature Medicine in September 2021) and report immunogenicity of a second BNT162b2 boost in participants who previously received homologous ChAdOx1 or heterologous ChAdOx1/BNT162b2 vaccination. Overall the work and writing is well-done and relevant, and merits publication. Relatively minor edits, as noted below, will improve the clarity and accuracy of the authors' messages.

The second paragraph of the main text remains relevant but would benefit from being updated to be more general and more "timeless," as the Omicron wave is declining significantly in most parts of the world.

Fig 1B vs. 1C reflects text in the 1st full paragraph on p. 3 that notes "anti-S IgG were comparable in both groups after the second BNT boost and were within the range of triple BNT vaccinated individuals." How did the analysis comparing anti-Spike IgG account for the longer interval (approximately 2 months longer) between the first and second injections in the CCB or CBB groups compared to the BBB groups?

Format of Figs 1B, C, and D: see comment on Fig 2.

Text, middle of p. 3 and Fig 1D: The authors note that spike-specific memory B-cells increase in both groups after the second boost—this seems to be a true but incomplete explanation of their findings. These spike-specific B-cells appear to be significantly less frequent ($p=0.19$) for C/B/B compared to C/C/B; this merits explanation in the text, given the ongoing dialogue and questions re: the best vaccine regimen(s) across immune compartments, and appreciating that the authors' CD4 and antibody (IgG and nAb) data seems to suggest that the C/B/B regimen may have advantages over the C/C/B regimen (e.g., achieve higher titers earlier).

Fig 2: This figure is full of interesting data, but it is not particularly well-conveyed in the current format. While the format is similar to that used in the Barros-Martins paper, its clarity is significantly diminished here for several reasons: the use of the x-axis to provide sample size at each timepoint; the addition of a third injection as currently depicted; and each timepoint denoted by its time after last vaccination (out of three vaccination timepoints). The figure would benefit from re-structuring to more clearly convey the vaccination and sampling timepoints:

- The timing of each vaccination is unclear as written here, with horizontal lines extending across the entire base of the graph for the 1st injections and so forth for the 2nd and 3rd injections; the format used to denote vaccinations in Figs 1B-D here or in the Barros-Martins ms seem more clear, or the authors might consider an arrow or an image of the syringe at the time of a given injection (perhaps different color syringes for ChAdOx1 than for BNT162b2 to distinguish them) or some other clearer manner of conveying the time of each injection compared to the timepoints on

the graph.

- In concert with a clear arrow or syringe or other designation of the time of each injection, using the x-axis to denote the time post-prime would be more straightforward to convey the timing of vaccination and sampling, beginning from time 0 at the prime/baseline on the far left of each graph section and moving to other clearly defined, relevant post-prime and post-boost timepoints to the right on the x-axis.

Text, 2nd and 3rd lines of p. 4: "Whilst initial ChAd/BNT immunization had induced neutralizing antibodies at high levels against all analyzed VoC..." The titers against Beta and Gamma variants do not appear to be particularly high-level after initial ChAd/BNT vaccination (even though as reported in Barros-Martins Extended Data Figure 6 they are higher than those elicited by homologous ChAd vaccination).

Text, 5th line of p. 4: Instead of "amounts mostly above those..." consider "titers mostly above those..."

Text on p. 4 appears to be repetitive: "More interestingly, heterologous BNT booster in the ChAd/ChAd vaccination group significantly raised numbers of spike-specific CD4+ T cells above amounts observed after the first boost. In contrast, individuals with heterologous ChAd/BNT vaccination only regained spike-specific CD4+ T cell levels corresponding to levels after the first boost (Fig. 2B)." And then later on p.4: "In contrast, the second BNT booster only made up for absent rise in spike-specific T cell responses after homologous ChAd/ChAd vaccination and merely restored spike-specific CD4+ T cell responses after heterologous vaccination."

Text near the bottom of p.4: "This suggests that variant-specific vaccines are required and need to be tested soon to better combat COVID-19 caused by Omicron and other emerging variants." This appears to be a significant over-statement of the data presented by the authors and I recommend revision. As noted by many now, neutralization even to the ancestral strain wanes over time, and the fact that neutralization is relatively preserved for four out of five VOCs evaluated here after vaccination with regimens based on the ancestral strain does not support a "requirement" for variant-specific vaccines. Furthermore, this begs the question: variant-specific vaccines are required for what? The placement of this sentence relative to the authors' observations re: nAb responses, specifically, suggests that the authors may have intended to convey that variant-specific vaccines could aid in eliciting higher and/or more durable nAb responses. That may aid in limiting acquisition of infection and may lead to faster viral clearance post-acquisition, but in the presence of the Spike-specific IgG, T- and B-cell responses that the authors (and others) have reported, it seems entirely unclear that variant-specific vaccines are required for prevention of severe illness, hospitalization or death (the authors mention this with respect to spike-specific T memory cells, and several real-world efficacy trials suggest that variant-specific vaccines are not required to prevent those endpoints).

Text on bottom of p.4: "These data confirm and extent..." – I believe the word "extent" should be "extend."

Fig 1B and C, and Extended Data Table 1: noting the different time points being compared across the BNT homologous and the ChAd homologous and ChAd/BNT heterologous regimens, the references to comparisons across these regimens would be improved if the authors adequately addressed the potentially significant impacts of different intervals and different sampling timepoints relative to prior vaccination. For example, the three-month difference in sampling timepoints after the 2nd vaccination (e.g., 4 vs. 7 months and 6 vs. 9 months) equates to approximately one half-life (~80-90d) previously published for nAb responses to at least the ancestral strain. Were these differences controlled for in any comparisons of immune responses between the B/B/B and the C/C/B or C/B/B regimens?

REVIEWER COMMENTS

Reviewer #1 (Remarks to the Author):

This manuscript investigated the effects of a 2nd booster (3rd dose) with BNT162b2 mRNA vaccine in individuals previously primed with two doses of the ChAd vaccine or a prime and one boost with the BNT mRNA vaccine. With the development of multiple COVID-19 vaccines and the need for worldwide distribution, vaccine mixing is inevitable and determining the ability of vaccine combinations to induce neutralizing antibody against new VOCs including Omicron is critical. This is a well-designed study comparing ChAd/ChAd/BNT to ChAd/BNT/BNT combinations and including a BNT only control and the methods which includes the analysis of both antibody and T cell responses and statistical methods are sound. However, there are some major and moderate issues that reduce confidence in the findings and interpretations.

Major:

Q1. A primary conclusion from this study is that an Omicron specific vaccine will be needed. This is based on data showing that after the 2nd booster dose, 31 or 47% of the vaccinees in both test groups failed to develop detectable neutralization activity against Omicron. A major concern with this conclusion is that the results in the control group (BNT/BNT/BNT) similarly show a relatively high rate of non-responders to Omicron after the 3rd dose but this is not consistent with previous reports which reported 100% of subjects developing neutralizing antibody after the 3rd dose in a BNT/BNT/BNT immunization regiment (<https://www.nature.com/articles/s41591-021-01676-0>). The authors should provide possible reasons for this discrepancy.

A1: *We agree to the reviewer that our data do not provide a formal evidence that an Omicron-specific vaccine will improve the lesser neutralizing immune activity observed after the third vaccination. This issue was also raised by reviewer #3. Indeed, recent publications suggest that Omicron-specific booster vaccination after standard vaccination leads to only limited differences in efficacy measured in mice (Ying B, et al. Cell, 28. March 2022, <https://doi.org/10.1016/j.cell.2022.03.037>.) and boosting non-human primates with either the mRNA-1273 or mRNA-Omicron vaccines elicit similar levels of protection upon challenge with SARS-CoV-2 Omicron (Gagne M, et al. Cell. 2022; 185(1):113-130.e15. doi: 10.1016/j.cell.2021.12.002.). We have therefore revised and extenuated our conclusion and included the above mentioned references (page 5).*

The study by Gruell H et al. demonstrates neutralizing activity assessed by a pseudovirus neutralization assays in 100% of study participants after the 3rd vaccination, consistent with other reports (Nemet I et al. N Engl J Med 2022, Liu L et al. Nature 2022). However, there are marked differences among these and other studies. Whilst Gruell H et al., Nemet et al., and Liu L et al. found no detectable neutralization against Omicron in 54% to 90% of samples shortly after the 2nd BNT vaccination, others found detectable neutralization against Omicron in 100% of samples at that time point (Schmidt N Engl J Med 2022, Wu M et al. Lancet 2022; Cameroni E et al. Nature 2022). Notably, Liu L et al. report that only 58% of boosted vaccinees (3 x BNT) mounted neutralization titers against authentic Omicron above the limit of detection, in keeping with the data from Wilhelm A et al. (MedRxiv 2022). These data highlight the challenge of cross-study comparisons when looking at serological assay results. Such differences are likely influenced by several factors and depend on the cells, authentic viruses, viral constructs or surrogate readouts used for assessing neutralization. We believe that the serological tests are primarily useful for comparing alterations captured by one test system over time but have clear limitations in cross-study comparisons.

We wish to emphasize another potential confounder. Studies about virus neutralization activity often select plasma samples numbers from larger cohorts, which are not always representative for definite vaccination schemes. Indeed, researcher may opt to handpick samples with high neutralizing activity in order to compare neutralization against different variants (Liu L et al. Nature 2022), which impedes on the representativeness of the respective study group. Similarly, Gruell et al. refer to the fact in their study “participants were relatively healthy which may be associated with a higher response to vaccinations...”. In our analysis all available samples of our ongoing observational cohort study were included, which we believe reflects a broader picture of the studied vaccine responses.

Finally, and to address a request of reviewer #2, we performed further validation and sample titrations of the sVNT. We now report that only 3-5% of individuals have no detectable neutralization against Omicron after the third vaccination (see revised Fig. 2A). We added additional data to the figure which show that 46% of HCP have neutralising activity against Omicron in the sVNT after the second vaccination of the heterologous vaccination scheme, which is in line with the literature mentioned above.

Q2. Immune responses measured after the 2nd booster are limited to only one time points at 2 weeks after the dose. A single measurement after the immunization reduces confidence in then conclusions. If the samples are available, the authors should include data at a later timepoint to validate the findings at day 14 and support their conclusion that “we expect the neutralization against Omicron to vanish rapidly.”

A2. *We agree to the reviewer that our data do not allow us to make firm conclusions about the persistence of neutralization against Omicron and we have weakened this statement accordingly in the revision. We are now referring to some recent reports about the antibody kinetics after the third vaccination suggesting waning immunity against Omicron after the third vaccination (Ferdinands JM et al. MMWR 2022; Andrews N et al. NEJM 2022; Chemaitelly H et al. Lancet 2022).*

Unfortunately, we are unable to provide a comprehensive follow up assessments of our HCP cohort. A significant proportion of HCP caught Omicron infection since our last examination in November 2021 and Omicron infections likewise impaired study team routine and performance. Several HCP meanwhile sought for a 4th vaccination with different mRNA vaccines at variable time points, because our university hospital does not provide a coordinated vaccination campaign for the 4th shot. Thus, we are unable to collect and assess blood samples in a representative way for formally addressing the reviewer’s request. A provisional assessment of the decline in anti-S IgG 2 weeks and 5 months after the third vaccination in a small subgroup of study participants is depicted in the figure below.

Figure: Anti-S IgG 2 weeks (2w) and 5 months (5m) after the third vaccination in vaccinees after heterologous (red) and homologous (blue) vaccination. The black dots represent participants with PCR-confirmed SARS-CoV-2 infection and newly detectable anti-NCP IgG after the last vaccination.

Q3. The discussion does not adequately compare the results from this study to similar studies that have already been published investigating the impact of homologous boosting with a 3rd dose of mRNA vaccine on breadth of neutralization against VOCs including Omicron. See link above and also: <https://www.nejm.org/doi/full/10.1056/NEJMc2119912>

A3. We thank the reviewer for referring to these timely studies (Gruell et al. Nat Med 2022, Pajon R et al. NEJM 2022). Meanwhile there have been several additional studies published examining the effect of the third (and forth) vaccination and we have considered these in the revised version of our manuscript.

Minor:

Q4. Some of the conclusions are based on % of subjects that developed detectable neutralizing antibody vs the VoCs including Omicron after the 2nd booster but these findings are not that clear from the figures. With regard to number of vaccinees that have detectable neutralizing antibody vs not, this could be improved if the authors insert a threshold or lower limit of detection line in each graph so it is more apparent which data points are above vs below. It would also help if the author indicated what % of subjects are above the threshold for each timepoint and group somewhere in the figure.

A4. As suggested by the reviewer, Fig. 2 and Suppl. Fig. 5 now contain a line representing the lower limit of detection so that it becomes clear which subjects are above the threshold for neutralizing antibody detection.

Q5. The authors established a virus neutralization test (sVNT) that positively correlates with neutralization detected via well-established pseudotyped virus neutralization test (pVNT). Does this

assay have a lower or higher limit of detection? This should be made clear in the methods or results section.

A5. *To determine the lower and higher limit of detection, we titrated the First WHO International Standard for anti-SARS-CoV-2 Immunoglobulin (NIBSC code 20/136) in the Wuhan sVNT. In general, the lower limit of detection is defined by the background signal of a pre-pandemic plasma pool (mean + 2SD, graphically displayed as shaded area in Suppl. Fig. 4). We defined the sVNT titer of a specific sample as the highest dilution that shows percent inhibition above the cut-off. Accordingly, the WHO International Standard 20/136 showed a titer of 1:1620 in the Wuhan sVNT (Suppl. Fig. 4, pink dot). The WHO arbitrarily assigned a value of 1000 neutralizing international units/mL (IU/mL) to the International Standard 20/136 ("Establishment of the WHO International Standard and Reference Panel for anti-SARS-CoV-2 antibody" (WHO reference number WHO/BS/2020.2402). Thus, the lower limit of detection is 1000 neutralizing IU/mL divided by 1620 = 0.62 neutralizing IU/mL. Further, the WHO Standard reaches saturation, i.e. 100% inhibition, at a dilution of 1:20 (Suppl. Fig. 4). Accordingly, the upper limit of detection is 1000 neutralizing IU/mL divided by 20 = 50 neutralizing IU/mL. This threshold sets the limit of detection for a particular dilution step. That means that samples with an end titer higher than 50 neutralizing IU/mL can still be adequately quantified by being tested in a higher dilution. Together, these data along with further experiments on the validation and characterization of the sVNT are now included in the Suppl. Text 1, Suppl. Fig.4, and Suppl. Table 3.*

Q6. In describing the results, it is not always clear if the authors are referring to the responses after the first booster or second booster. It would be helpful if the same terms (i.e. after first booster vs after second booster) are consistently used throughout the manuscript.

A6. *We agree to the reviewer that the terminology was inconsistent. We have revised this and now refer to the 2nd vaccination (corresponding to the 1st booster) and 3rd vaccination (corresponding to the 2nd booster) throughout the manuscript.*

Q7. In the methods, the authors should include information on the virus specificity (Wu-1?) of the peptide pools they used for stimulation in the T cell assays.

A7. *We thank the reviewer for this suggestion. We used peptide pools of the Wuhan variant spike protein and included this information in the M&M section of the revised manuscript.*

Reviewer #2 (Remarks to the Author):

The authors summarized longitudinal immune responses after ChAd/ChAd and ChAd/BNT, as well as immune responses 14 days after a BNT boost following the primary series against multiple variants of SARS-CoV-2. The paper is very clearly written and easy to follow. Here are few comments.

Major:

Q8. The conclusion about the sVNT assay being a reliable tool to quantitatively assess neutralization seems weak. Such statement would need to have a formal validation of the assay including parameters such as accuracy, precision, range, limits of detection or quantification etc. Why an R² value of 0.7044 supports this?

A8. *According to the reviewer's suggestion, we performed further experiments for the formal validation of the sVNT assay. All these information can be found in the Suppl. Text 1, Suppl. Fig.4, and Suppl. Table 3.*

In brief, we assessed the precision and within-run coefficient of variation (CV) and confirmed that the large majority of sVNT measures corresponded to the recommendations of the EMA guideline for bioanalytical method validation that applies for the highly regulated areas of animal toxicokinetic studies and all phases of clinical trials. [According to the EMA guideline for bioanalytical method validation, both within-run and between-run CV values should not exceed 15% for high and medium quality control samples, except for the samples at lower limit of quantification, for which CVs should not exceed 20% (EMEA/CHMP/EWP/192217/2009 Rev. 1 Corr. 2 accessed via <https://www.ema.europa.eu/en/bioanalytical-method-validation> on April 28, 2022).]

Second, we assessed the First WHO International Standard for anti-SARS-CoV-2 Immunoglobulin (NIBSC code 20/136) in the Wuhan sVNT and defined the lower limit of detection. We determined the percent inhibition of the First WHO International Standard and demonstrate that our result is close to the percent inhibition of commercial Genscript sVNT when used under similar conditions [“Establishment of the WHO International Standard and Reference Panel for anti-SARS-CoV-2 antibody” (WHO reference number WHO/BS/2020.2402)]. Unfortunately, since the First WHO International Standard for anti-SARS-CoV-2 Immunoglobulin (NIBSC code 20/136) is no longer available, we could perform these tests only for the Wuhan sVNT but not for other VOC sVNTs. Finally, we determined the specificity by assessing 40 different pre-pandemic plasma samples in the sVNT.

Q9. Given that the correlates of the Moderna vaccine have been reported (Gilbert et al. 2022), it’s ideal if the magnitudes of the reported neutralization titers could be converted to the WHO international units for a comparison to other vaccine regimens.

A9. *The data by Gilbert et al. help defining immune marker correlates of protection and may guide approval decisions for COVID-19 vaccines. The authors measured neutralizing and binding antibodies as correlates of protection at the time of second vaccination and 4 weeks later, with values reported in standardized World Health Organization international units. The latter was achieved by calibration with the First WHO International Standard for Anti-SARS-CoV-2 Immunoglobulin (20/136).*

Moreover, and as the authors point out, an important limitation the Gilbert study is the fact that all COVID-19 cases resulted from infections with viruses with a spike sequence similar to that of the vaccine strain, which precluded the assessment of robustness of correlates to SARS CoV-2 variants of concern. Thus, calibration of neutralization assays to standardize neutralizing activity against Delta or Omicron has not been performed yet.

Despite these limitations, we converted the results from the anti-Spike IgG assay into BAU/mL according to the manufactures description, calibrated the sVNT with the First WHO International Standard for Anti-SARS-CoV-2 Immunoglobulin (20/136) and revised the figures accordingly.

Q10. Have the authors tried to adjust for any potential confounding factors in the comparisons between the two groups given that they were not randomized groups. This will be helpful at least as a supportive analysis.

A10. *This is valid point and we have included more data to address and illustrate this issue in the Suppl. Fig. 1. Our study population size and set of variables is too small for formal statistical approach to explore potential confounders by e.g. a regression analysis. Instead, and like in the analysis for the Barros-Martins et al. report, we looked for two variables, which are likely factors influencing the study outcomes, namely age and sex. As shown in Suppl. Fig. 1 of the revised manuscript, we depicted anti-S IgG results in relation to age and sex. Anti-S IgG concentrations of male and females after the third*

vaccination were not statistically significant different (Suppl. Fig. 1a). Interestingly, we observed a small but significant correlation of age and anti-S IgG in the heterologous vaccinated group, but surprisingly with higher anti-S IgG at increasing age (Suppl. Fig. 1c). This was absent in the group with initially homologous vaccination. In summary, whilst we cannot exclude all potential confounders affecting our observation of similar anti-S IgG results after the third vaccination, the revision of the Suppl. Data now including the above mentioned results provides the reader with a more detailed description of our data.

Minor:

Q11. The word “efficacy” in “Assessing the *efficacy* of a second boost vaccination” isn’t appropriate as no clinical outcomes are assessed.

A11. We have replaced the word “efficacy” by the word “effects”.

Reviewer #3 (Remarks to the Author):

Thank you for the opportunity to review. In this manuscript, the authors follow up their prior work (Barros-Martins, et al in Nature Medicine in September 2021) and report immunogenicity of a second BNT162b2 boost in participants who previously received homologous ChAdOx1 or heterologous ChAdOx1/BNT162b2 vaccination. Overall the work and writing is well-done and relevant, and merits publication. Relatively minor edits, as noted below, will improve the clarity and accuracy of the authors’ messages.

Q12. The second paragraph of the main text remains relevant but would benefit from being updated to be more general and more “timeless,” as the Omicron wave is declining significantly in most parts of the world.

A12. We agree and have revised accordingly to consider the most recent developments since our initial submission.

Q13. Fig 1B vs. 1C reflects text in the 1st full paragraph on p. 3 that notes “anti-S IgG were comparable in both groups after the second BNT boost and were within the range of triple BNT vaccinated individuals.” How did the analysis comparing anti-Spike IgG account for the longer interval (approximately 2 months longer) between the first and second injections in the CCB or CBB groups compared to the BBB groups?

A13. We have addressed this point further below and added the following sentence to the revised manuscript (page 3): “Please note that in the BNT/BNT/BNT group samples were collected at different time points than in the other two groups.”

Q14. Format of Figs 1B, C, and D: see comment on Fig 2.

A14. We have addressed this point further below together with comments to Fig. 2 (see Q16)

Q15. Text, middle of p. 3 and Fig 1D: The authors note that spike-specific memory B-cells increase in both groups after the second boost—this seems to be a true but incomplete explanation of their findings. These spike-specific B-cells appear to be significantly less frequent ($p=0.19$) for C/B/B compared to C/C/B; this merits explanation in the text, given the ongoing dialogue and questions re: the best vaccine regimen(s) across immune compartments, and appreciating that the authors’ CD4 and

antibody (IgG and nAb) data seems to suggest that the C/B/B regimen may have advantages over the C/C/B regimen (e.g., achieve higher titers earlier).

A15. *We have added a sentence describing the differences as suggested (page 3). “Interestingly, 2 weeks after a third immunization with BNT, spike-specific memory B were significantly higher in the ChAd/ChAd as compared to the ChAd/BNT prime boost group.”*

Q16. Fig 2: This figure is full of interesting data, but it is not particularly well-conveyed in the current format. While the format is similar to that used in the Barros-Martins paper, its clarity is significantly diminished here for several reasons: the use of the x-axis to provide sample size at each timepoint; the addition of a third injection as currently depicted; and each timepoint denoted by its time after last vaccination (out of three vaccination timepoints). The figure would benefit from re-structuring to more clearly convey the vaccination and sampling timepoints:

- The timing of each vaccination is unclear as written here, with horizontal lines extending across the entire base of the graph for the 1st injections and so forth for the 2nd and 3rd injections; the format used to denote vaccinations in Figs 1B-D here or in the Barros-Martins ms seem more clear, or the authors might consider an arrow or an image of the syringe at the time of a given injection (perhaps different color syringes for ChAdOx1 than for BNT162b2 to distinguish them) or some other clearer manner of conveying the time of each injection compared to the timepoints on the graph.
- In concert with a clear arrow or syringe or other designation of the time of each injection, using the x-axis to denote the time post-prime would be more straightforward to convey the timing of vaccination and sampling, beginning from time 0 at the prime/baseline on the far left of each graph section and moving to other clearly defined, relevant post-prime and post-boost timepoints to the right on the x-axis.

A16. *We agree to the reviewer and made several revisions according to his suggestions. In addition, we now mention and depict that additional vaccinees after prime boost vaccination were included in this analysis, which were not yet assessed for our previous interim report (Barros-Martins 2021) and we have highlighted these data in the figures in colors. Importantly, these additional study participants, which were not considered in the analysis in the Barros-Martins paper, did not change any of the differences reported previously. Finally, we have removed all sample size numbers from the data figures and provide this information now separately in the Suppl. Table 2.*

Q17. Text, 2nd and 3rd lines of p. 4: “Whilst initial ChAd/BNT immunization had induced neutralizing antibodies at high levels against all analyzed VoC...” The titers against Beta and Gamma variants do not appear to be particularly high-level after initial ChAd/BNT vaccination (even though as reported in Barros-Martins Extended Data Figure 6 they are higher than those elicited by homologous ChAd vaccination).

A17. *We agree to the reviewer that our description was not adequately reporting the data and thus have revised this section in the manuscript. “Whilst initial ChAd/BNT immunization had induced neutralizing antibodies at high levels against all analyzed VoC except for Beta and Gamma after ChAd/ChAd vaccination, the following decline was more than restored by the 3rd immunization (Fig. 2A).”*

Q18. Text, 5th line of p. 4: Instead of “amounts mostly above those...” consider “titers mostly above those...”

A18. *We have changed this as suggested.*

Q19. Text on p. 4 appears to be repetitive: “More interestingly, heterologous BNT booster in the ChAd/ChAd vaccination group significantly raised numbers of spike-specific CD4+ T cells above amounts observed after the first boost. In contrast, individuals with heterologous ChAd/BNT vaccination only regained spike-specific CD4+ T cell levels corresponding to levels after the first boost (Fig. 2B).” And then later on p.4: “In contrast, the second BNT booster only made up for absent rise in spike-specific T cell responses after homologous ChAd/ChAd vaccination and merely restored spike-specific CD4+ T cell responses after heterologous vaccination.”

A19. *We decided to leave the first sentence describing the results and changed the second as part of the discussion. (“The 3rd BNT immunization made up for absent rise in spike-specific T cell responses after homologous ChAd/ChAd vaccination. These data confirm and extend reports that ChAd does not boost cellular responses after ChAd/ChAd vaccination”)*

Q20. Text near the bottom of p.4: “This suggests that variant-specific vaccines are required and need to be tested soon to better combat COVID-19 caused by Omicron and other emerging variants.” This appears to be a significant over-statement of the data presented by the authors and I recommend revision. As noted by many now, neutralization even to the ancestral strain wanes over time, and the fact that neutralization is relatively preserved for four out of five VOCs evaluated here after vaccination with regimens based on the ancestral strain does not support a “requirement” for variant-specific vaccines. Furthermore, this begs the question: variant-specific vaccines are required for what? The placement of this sentence relative to the authors’ observations re: nAb responses, specifically, suggests that the authors may have intended to convey that variant-specific vaccines could aid in eliciting higher and/or more durable nAb responses. That may aid in limiting acquisition of infection and may lead to faster viral clearance post-acquisition, but in the presence of the Spike-specific IgG, T- and B-cell responses that the authors (and others) have reported, it seems entirely unclear that variant-specific vaccines are required for prevention of severe illness, hospitalization or death (the authors mention this with respect to spike-specific T memory cells, and several real-world efficacy trials suggest that variant-specific vaccines are not required to prevent those endpoints).

A20. *We agree to the reviewer that this sentence appears exaggerated and we have changed the interpretation and wording according to the reviewer’s suggestion. We now refer to two recent publications providing experimental data on this issue. “Studies about Omicron-specific 3rd vaccination after two standard vaccination lead to only limited differences in efficacy measured in mice (Ying et al Cell 2022) and a third vaccination of non-human primates with either the mRNA-1273 or mRNA-Omicron vaccines elicit similar levels of protection upon challenge with SARS-CoV-2 Omicron (Gagne M et al. Cell 2022).”*

Q21. Text on bottom of p.4: “These data confirm and extent...” – I believe the word “extent” should be “extend.”

A21. *Changed as suggested.*

Q22. Fig 1B and C, and Extended Data Table 1: noting the different time points being compared across the BNT homologous and the ChAd homologous and ChAd/BNT heterologous regimens, the references to comparisons across these regimens would be improved if the authors adequately addressed the potentially significant impacts of different intervals and different sampling timepoints relative to prior vaccination. For example, the three-month difference in sampling timepoints after the 2nd vaccination (e.g., 4 vs. 7 months and 6 vs. 9 months) equates to approximately one half-life (~80-90d) previously

published for nAb responses to at least the ancestral strain. Were these differences controlled for in any comparisons of immune responses between the B/B/B and the C/C/B or C/B/B regimens?

A22. *We fully agree to the reviewer's comment. This is the reason why we have not performed any statistical comparison of the triple BNT vaccinated individuals and also separated the group in Suppl. Table 1. We consider this group as an independent control group, since it is impossible to adequately control for the different blood collection time points after vaccination. Still, we think for illustrating the within group effect of the 3rd vaccination, the "BNT only" group provides some valuable information. For better explanation, we revised Fig. 1A, which contains the participant recruitment and vaccination and blood-sampling scheme. We also added a sentence highlighting this issue on page 4. "Please note that in the BNT/BNT/BNT group samples were collected at different time points than in the other two groups."*

REVIEWERS' COMMENTS

Reviewer #1 (Remarks to the Author):

The authors revisions in response to the critiques were appropriate and addressed all the major concerns.

Reviewer #2 (Remarks to the Author):

-- in the description of the validation experiments, please clarify:

- 1) how many "different" samples were used and how they were shown in Fig S4.
- 2) it seems like the %CV was only calculated for intra-assay variability based on 6 replicates of the same sample, not variability covering multiple days or multiple operators. So, at the minimal, please clarify the level of precision assessment and the experiments are not "full validation".

-- it would be helpful to comment on the antibody levels relative to what have been observed in other studies, especially on the converted scale.

Reviewer #3 (Remarks to the Author):

I've re-reviewed the revised manuscript by Behrens, et al, as well as the responses to our prior reviews. I appreciate the edits the authors have made and have no further substantive comments. The authors have adequately addressed my concerns and I support publication of this timely and useful work.

Point by point response

Reviewer #2

- 1) how many "different" samples were used and how they were shown in Fig S4.

Response: We are sorry, but apparently our mistakenly description in the text (Suppl. Note 1 in the revised manuscript), which referred to Suppl. Fig. S4 (Suppl. Fig.4 of the revised manuscript), caused confusion. This figure depicts the titration of the First WHO International Standard for anti-SARS-CoV-2 Immunoglobulin (NIBSC code 20/136) only. Thus, these data points are based on a single measurement.

- 2) it seems like the %CV was only calculated for intra-assay variability based on 6 replicates of the same sample, not variability covering multiple days or multiple operators. So, at the minimal, please clarify the level of precision assessment and the experiments are not "full validation".

Response: Again, it seem that we caused confusion due to a mislabeling. In detail, we analyzed seventeen different plasma samples covering a wide range of neutralizing titers. We tested these seventeen different samples in three independent sVNT runs to assess the between-run CV with six replicates of each sample to determine the within-run CV. Each run was performed on a different day. All results are reported in Suppl. Note 1.

- 3) it would be helpful to comment on the antibody levels relative to what have been observed in other studies, especially on the converted scale.

Response: We obtained mean antibody levels after initial homologous ChAd/ChAd or heterologous ChAd/BNT vaccination followed by a third BNT vaccination of about 6.000 BAU/mL or after triple BNT vaccination of about 4.000 BAU/mL.

Wratil PR et al. (Nat Med 2022) found anti-S levels of 4.000 to 6.000 BAU/mL in triple vaccinated individuals or convalescent persons with double vaccination after COVID-19 (same ELISA as ours). Similarly, van Gils MJ et al. (PLoS Med 2022) reported about 2.000-3.000 BAU/mL after triple BNT vaccination (different ELISA). Finally, Hayashi YJ et al. (J Infect, 2022) obtained anti-S IgG levels of about 3.000 BAU/mL in individuals after ChAd/ChAd/BNT vaccination (different ELISA). Considering differences in time intervals in between vaccinations, our results are well within the range of other reports. We have included this briefly at the beginning of the revised discussion.